# EduQate: Generating Adaptive Curricula through RMABs in Education Settings

## Abstract

There has been significant interest in the development of personalized and adaptive educational tools that cater to a student's individual learning progress. A crucial aspect in developing such tools is in exploring how mastery can be achieved across a diverse yet related range of content in an efficient manner. While Reinforcement Learning and Multi-armed Bandits have shown promise in educational settings, existing works often assume the independence of learning content, neglecting the prevalent interdependencies between such content. In response, we introduce *Education Network Restless Multi-armed Bandits* (EdNetRMABs), utilizing a network to represent the relationships between interdependent arms. Subsequently, we propose *EduQate*, a method employing interdependency-aware Q-learning to make informed decisions on arm selection at each time step. We establish the optimality guarantee of EduQate and demonstrate its efficacy compared to baseline policies, using students modeled from both synthetic and real-world data.

## 1 Introduction

The COVID-19 pandemic has accelerated the adoption of educational technologies, especially on eLearning platforms. Despite abundant data and advancements in modeling student learning, effectively capturing the learning process with interdependent content remains a significant challenge [9]. The conventional rules-based approach to creating personalized learning curricula is impractical due to its labor-intensive nature and need for expert knowledge. Machine learning-based systems offer a scalable alternative, automatically generating personalized content to optimize learning [22, 24].

One possible approach to model the learning process is the Restless Multi-Armed Bandits (RMAB, [26]), where a teacher agent selects a subset of arms (concepts) to teach each round. However, RMAB's assumption that arms are independent is unrealistic in educational settings. For example, solving a math question on the area of a triangle requires knowledge of algebra, arithmetic, and geometry. Practicing this question should enhance proficiency in all three areas. Models that ignore such interdependencies may inaccurately predict knowledge levels by assuming each exercise impacts only a single area.

In response to this challenge, we introduce an interdependency-aware RMAB model to the education setting. We posit that by acknowledging and modeling the learning dynamics of interdependent content, both teachers and algorithms can strategically leverage overlapping utility to foster mastery over a broader range of topics within a curriculum. We advocate for RMABs as a fitting model for this context, as the inherent dynamics of such a model align closely with the learning process.

In this study, our objective is to derive a teacher policy that effectively recommends educational content to students, accounting for interdependencies among the content to enhance overall utility (that characterizes understanding and retention of content). Our contributions are as follows:

1. We introduce Restless Multi-armed Bandits for Education (EdNetRMABs), enabling the modeling of learning processes with interdependent educational content.

2. We propose EduQate, a Whittle index-based heuristic algorithm that uses Q-learning to compute an inter-dependency-aware teacher policy. Unlike previous methods, EduQate does not require knowledge of the transition matrix to compute an optimal policy.

3. We provide a theoretical analysis of EduQate, demonstrating guarantees of optimality.

4. We present empirical results on simulated students and real-world datasets, showing the effectiveness of EduQate over other teacher policies.

## 2 Related Work and Preliminaries

### 2.1 Restless Multi-Armed Bandits

The selection of the right time and manner for limited interventions is a problem of great practical importance across various domains, including health intervention [17, 5], anti-poaching operations [20], education [13, 6, 2], etc. These problems share a common characteristic of having multiple arms in a Multi-armed Bandit (MAB) problem, representing entities such as patients, regions of a forest, or students' mastery of concepts. These arms evolve in an uncertain manner, and interventions are required to guide them from "bad" states to "good" states. The inherent challenge lies in the limited number of interventions, dictated by the limited resources (e.g., public health workers, the number of student interactions). RMAB, a generalization of MAB, offers an ideal model for representing the aforementioned problems of interest. RMAB allows non-active bandits to also undergo the Markovian state transition, effectively capturing uncertainty in arm state transitions (reflecting uncertain state evolution), actions (representing interventions), and budget constraints (illustrating limited resources).

RMABs and the associated Markov Decision Processes (MDP) for each arm offer a valuable model for representing the learning process. Firstly, leveraging the MDPs associated with each arm provides the flexibility to adopt nuanced modeling of learning content, accommodating different learning curves for various content based on students' strengths and weaknesses. Secondly, the transition probabilities serve as a useful mechanism to model forgetting (through state decay due to passivity or negligence) and learning (through state transitions to the positive state from repeated practice). Considering these aspects, RMABs prove to be a beneficial framework for personalizing and generating adaptive curricula across a diverse range of students.

In general, computing the optimal policy for a given set of restless arms in RMABs is recognized as a PSPACE-hard problem [18]. The Whittle index [26] provides an approach with a tractable solution that is provably optimal, especially when each arm is indexable. However, proving indexability can be challenging and often requires specification of the problem's structure, such as the optimality of threshold policies [17, 16]. Moreover, much of the research on Whittle Index policies has focused on two-action settings or requires prior knowledge of the transition matrix of the RMABs. Meeting these conditions proves challenging in the educational context, where diverse students interact with educational systems, each possessing different prior knowledge and distinct learning curves for various topics.

WIQL [5], on the other hand, employs a Q-learning-based method to estimate the Whittle Index and has demonstrated provable optimality without requiring prior knowledge of the transition matrix. We utilize WIQL as a baseline method in our subsequent experiments.

In a recent investigation by [12], RMABs were explored within a network framework, requiring the agent to manage a budget while allocating a high-cost, high-benefit resource to one arm to "unlock" potential lower-cost, intermediate-benefit resources for the arm's neighbors. The network effects emphasized in their work are triggered by an intentional, active action, enabling the agent to choose to propagate positive externalities to a selected arm's neighbors within budget constraints. In contrast, our study delves into scenarios where network effects are indirect results of an active action, and the agent lacks direct control over such effects. Thus, the challenge lies in accurately modeling these network effects and leveraging them when beneficial.

## 2.2 Reinforcement Learning in Education

In the realm of education, numerous researchers have explored optimizing the sequencing of instructional activities and content, assuming that optimal sequencing can significantly impact student learning. RL is a natural approach for making sequential decisions under uncertainty [1]. While RL has seen success in various educational applications, effectively sequencing interdependent content in a personalized and adaptive manner has yielded mixed or insignificant results compared to baseline teacher policies [11, 21, 8]. In general, these RL works focus on data-driven methods using student activity logs to estimate students' knowledge states and progress, assuming that the interdependencies between learning content are encapsulated in students' learning histories [9, 3, 19]. In contrast, our work focuses on modelling these interdependencies directly.

Of particular relevance are factored MDPs applied to skill acquisition introduced by [11]. While factored MDPs account for interdependencies amongst skills, decentralized policy learning is infeasible as policies must consider the joint state space. Our work leverages the advantage of decentralized policy learning provided by RMABs and introduces a novel decentralized learning approach that exploits interdependencies between arms.

Complementary to RL methods in education is the utilization of knowledge graphs to uncover relationships between learning content [9]. Existing research primarily focuses on establishing these relationships through data-driven methods (e.g. [7, 23]) often leveraging student-activity logs. In this work, we complement such research by presenting an approach where bandit methods can effectively operate with knowledge graphs derived by such methods.

# 3 Model

In this section, we introduce the Restless Multi-Armed Bandits for Education (EdNetRMABs). It is important to note that while we specifically apply EdNetRMABs to the education setting, the framework can be seamlessly translated to other scenarios where modeling the effects of active actions within a network is critical. For ease of access, a table of notations is provided in Table 2.

In education, a teacher recommends learning content, or items, to maximize student education, often with content from online platforms. Items are grouped by topics, such as "Geometry," where exposure to one piece of content can enhance knowledge across others in the same group. This cumulative learning effect which we refer to as "network effects", implies that exposure to an item is likely to positively impact the student's success on items within the same group. A successful teacher accurately estimates a student's knowledge state over repeated interactions, leveraging these network effects to promote both breadth and depth of understanding through recommendations.

## 3.1 EdNetRMABs

The RMAB model tasks an agent with selecting $k$ arms from $N$ arms, constrained by a limit on the number of arms that can be pulled at each time step. The objective is to find a policy that maximizes the total expected discounted reward, assuming that the state of each arm evolves independently according to an underlying MDP.

The EdNetRMABs model extends RMABs by allowing for active actions to propagate to other arms dependent on the current arm when it is being pulled, thus relaxing the assumption of independent arms. This is operationalized by organising the arms in a network, and pulling of an arm results in changes for its neighbors, or members in the same group.

When applied to education setting, the EdNetRMABs is formalized as follows:

**Arms** Each arm, denoted as $i \in 1, ..., N$, signifies an item. In the context of this networked environment, each arm belongs to a group $\phi \in \{1, ..., L\}$ representing the overarching topic that encompasses related items. It's important to note that arm membership is not mutually exclusive, allowing arms to be part of multiple groups. This flexibility enables a more nuanced modeling of interdependencies among educational content. For instance, a question involving the calculation of the area of a triangle may span both arithmetic and geometry groups.

**State space**  In this framework, each arm possesses a binary latent state, denoted as $s_i \in \{0, 1\}$, where "0" represents an "unlearned" state, and "1" indicates a "learned" state. Considering all arms collectively, these states serve as a representation of the student's overall knowledge state. In the current work, it is assumed that the states of all arms are fully observable, providing a comprehensive model of the student's understanding of the various educational concepts.

**Action space**  To capture the network effects associated with arm pulls, we depart from the conventional RMAB framework with a binary action space $A = \{0, 1\}$ by introducing a pseudo-action. In this modified setup, the action space is extended to $A = \{0, 1, 2\}$, where actions 0 and 2 represent "no-pull" and "pull", as commonly used in bandit literature. Notably, in EdNetRMABs, a third action 1 is introduced to simulate the network effects resulting from pulling another arm within the same group. It is important to clarify that agents do not directly engage with action 1 but we employ it solely for modeling network effects, hence the term "pseudo-action".

**Transition function**  For a given arm $i$, let $P_{s,s'}^{a,i}$ represent the probability of the arm transitioning from state $s$ to $s'$ under action $a$. It's noteworthy that, in typical real-world educational settings, the actual transition functions governing the states of the arms are often unknown and, even for the same concept, may vary among students due to differences in prior knowledge. To address this challenge, we adopt model-free approaches in this study, devising methods to compute the teacher policy without relying on explicit knowledge of these transition functions. In the following experiments, we maintain the assumption of non-zero transition probabilities, and enforce constraints that are aligned with the current domain [17]: (i) The arms are more likely to stay in the positive state than change to the negative state: $P_{0,1}^0 < P_{1,1}^0$, $P_{0,1}^1 < P_{1,1}^1$ and $P_{0,1}^2 < P_{1,1}^2$; (ii) The arm tends to improve the latent state if more efforts is spent on that arm, i.e., it is active or semi-active: $P_{0,1}^0 < P_{0,1}^1 < P_{0,1}^2$ and $P_{1,1}^0 < P_{1,1}^1 < P_{1,1}^2$.

With the formalization of the EdNetRMABs model provided, we now apply it to an educational context. In this scenario, the agent assumes the role of a teacher and takes actions during each time step $t \in \{1, ..., T\}$. Specifically, at each time step, the teacher recommends an item for the student to study. We represent the vector of actions taken by the teacher at time step $t$ as $\mathbf{a}^t \in \{0, 1, 2\}^N$. Here, arm $i$ is considered to be active at time $t$ if $a^t(i) = 2$ and passive when $a^t(i) = 0$. When arm $i$ is pulled, the set of arms that share the same group membership as arm $i$, denoted as $\phi_i^-$ under goes the pseudo-action, represented as $a^t(j) = 1$ for all $j \in \phi^-$. In our framework, the teacher agent acts on exactly one arm per time step to simulate the real-world constraint that the teacher can only recommend one concept to students ( $\sum_i I_{a^t(i)=2} = 1, \forall t$ ). Subsequent to taking action, the teacher receives $\mathbf{s}^t \in \{0, 1\}^N$, a vector reflecting the state of all arms, and reward $r_t = \sum_{i=1}^N s^t(i)$. The vector $\mathbf{s}^t$ represents the overall knowledge state of the student. The teacher agent's goal, therefore, is to maximize the long term rewards, either discounted or averaged.

# 4  EduQate

Q-learning [25] is a popular reinforcement learning method that enables an agent to learn optimal actions in an environment by iteratively updating its estimate of state-action value, $Q(s, a)$, based on the rewards it receives. At each time step $t$, the agent takes an action $a$ using its current estimate of $Q$ values and current state $s$, thus received a reward of $r(s)$ and new state $s'$. We provide an abridged introduction to Q-learning in the Appendix F.

Expanding upon Q-learning, we introduce *EduQate*, a tailored Q-learning approach designed for learning Whittle-index policies in EdNetRMABs. In the interaction with the environment, the agent chooses a single item, represented by arm $i$, to recommend to the student. In this context, the agent possesses knowledge of the group membership $\phi_i$ of the selected arm and observes the rewards generated by activating arm $i$ and semi-activating arms in $\phi_i^-$. EduQate utilizes this interaction to learn the Q-values for all arms and actions.

To adapt Q-learning to EdNetRMABs, we propose leveraging the learned Q-values to select the arm with the highest estimate of the Whittle index, defined as:

**Algorithm 1** Q-Learning for EdNetRMABs (EduQate)

---
**Input:** Number of arms $N$
Initialize $Q_i(s, a) \leftarrow 0$ and $\lambda_i(s) \leftarrow 0$ for each state $s \in S$ and each action $a \in \{0, 1, 2\}$, for each arm $i \in 1, ..., N$.
Initialize replay buffer $D$ with capacity $C$.
**for** $t$ in $1, ..., T$ **do**
  $\epsilon \leftarrow \frac{N}{N+t}$
  With probability $\epsilon$, select one arm uniformly at random. Otherwise, select arm with highest Whittle Index, $i = \arg\max_i \lambda_i$.
  **for** arm $n$ in $1, ..., N$ **do**
    **if** $n \neq i$ **then**
      Set arm $n$ to passive, $a_n^t = 0$
    **else**
      Set arm $n$ to active, $a_n^t = 2$
      **for** $j \in \phi_i^-$ **do**
        Set arms in same group as $i$ to semi-active, $a_j^t = 1$
      **end for**
    **end if**
  **end for**
  Execute actions $\mathbf{a^t}$ and observe reward $r^t$ and next state $s^{t+1}$ for all arms
  Store experience $(s^t, \mathbf{a^t}, \mathbf{r^t}, \mathbf{s^{t+1}})$ in replay buffer $D$.
  Sample minibatch $B$ of Experience from replay buffer $D$.
  **for** Experience in minibatch $B$ **do**
    Update $Q_n(s, a)$ using Q-learning update in Equation 11.
    Compute $\lambda_n$ using Equation 1
  **end for**
**end for**

---

$$\lambda_i = Q(s_i, a_i = 2) - Q(s_i, a_i = 0) + \sum_{j \in \phi_i^-} (Q(s_j, a_j = 1) - Q(s_j, a_j = 0)) \tag{1}$$

Here, $\lambda_i$ is the Whittle Index estimate for arm $i$. In essence, the Whittle Index of arm $i$ is computed as the linear combination of the value associated with taking action on arm $i$ over passivity and the value of associated with semi-actively engaging with members from same group, compared to passivity.

To improve the convergence of Q-learning, we incorporate Experience Replay [15]. This involves saving the teacher algorithm's previous experiences in a replay buffer and drawing mini-batches of samples from this buffer during updates to enhance convergence. In Section 4.1, we prove that EduQate will converge to the optimal policy. However, in practice, we may not have enough episodes to fully train EduQate. Therefore, we propose Experience Replay to mitigate the cold-start problem common in RL applications, a common problem where initial student interactions with sub-optimal teachers can lead to poor learning experiences [3].

The pseudo-code is provided in Algorithm 1. Similar to WIQL [5], we employ a $\epsilon$-decay policy that facilitates exploration and learning in the early steps, and proceeds to exploit the learned Q-values in later stages.

## 4.1 Analysis of EduQate

In this section, we analyze EduQate closely, and show that EduQate does not alter the optimality guarantees of Q-learning under the constraint that k = 1 (Theorem 1). Our method relies on the assumption that teachers are limited to assign 1 item to the student at each time step. Theorem 2 analyzes EduQate under the conditions that $k > 1$. Since our setting involves the semi-active actions, we should compute Equation 1. To reiterate, $\phi_i$ here refers to the group that arm $i$ belongs to, and $\phi_i^-$ is the same group but does not include arm $i$. If arm $i$ is selected, then all the remaining arms in group $\phi_i^-$ should be semi-active.

**Theorem 1** *Choosing the top arm with the largest $\lambda$ value in Equation 1 is equivalent to maximizing the cumulative long-term reward.*

*Proof.* According to the approach, we select the arm according to the $\lambda$ value. Assume arm $i$ has the highest $\lambda$ value, then for any arm $j$ where $j \neq i$, we have

$$\lambda_i \geq \lambda_j \tag{2}$$

According to the definition of $\lambda$ in Equation 1, we move the negative part to the other side, and the left side becomes:

$$Q(s_i, a_i = 1) + \sum_{i \in \phi_i^-} (Q(s_i, a_i = 1)) + Q(s_j, a_j = 0) + \sum_{j \in \phi_j^-} (Q(s_j, a_j = 0))$$

and the right side is similar. There are three cases:

- arm $i$ and arm $j$ are not connected, and group $\phi_i$ and $\phi_j$ has no overlap, i.e., $\phi_i \cap \phi_j = \emptyset$. We add $\sum_{z \notin \phi_i \wedge z \notin \phi_j} Q(s_z, a_z = 0)$ on both sides. This denotes the addition of $Q(s_z, a_z = 0)$ for all arm $z$ that are not included in the set of $\phi_i$ or $\phi_j$. We have the left side:

$$Q(s_i, a_i = 1) + \sum_{i \in \phi_i^-} (Q(s_i, a_i = 1)) + Q(s_j, a_j = 0) + \sum_{j \in \phi_j^-} (Q(s_j, a_j = 0)) + \sum_{z \notin \phi_i \wedge z \notin \phi_j} Q(s_z, a_z = 0)$$

$$= Q(s_i, a_i = 1) + \sum_{i \in \phi_i^-} (Q(s_i, a_i = 1)) + \sum_{j \notin \phi_i} (Q(s_j, a_j = 0))$$

$$= Q(\mathbf{s}, \mathbf{a} = \mathbb{I}_i) \tag{3}$$

Similarly, we do the same for the right side and thus, the equation 2 becomes

$$Q(\mathbf{s}, \mathbf{a} = \mathbb{I}_i) \geq Q(\mathbf{s}, \mathbf{a} = \mathbb{I}_j)$$

- arm $i$ and arm $j$ are not connected, but group $\phi_i$ and $\phi_j$ has overlap, i.e., $\phi_i \cap \phi_j \neq \emptyset$. In this case, we add $\sum_{z \notin \phi_i \wedge z \notin \phi_j} Q(s_z, a_z = 0) - \sum_{z \in \phi_i \cap \phi_j} Q(s_z, a_z = 0)$ on both sides.

- arm $i$ and arm $j$ are connected, and group $\phi_i$ and $\phi_j$ has overlap, i.e., $\phi_i \cap \phi_j \neq \emptyset$, and $\{i, j\} \subset \phi_i \cap \phi_j$. This case is similar to the previous one, we add $\sum_{z \notin \phi_i \wedge z \notin \phi_j} Q(s_z, a_z = 0) - \sum_{z \in \phi_i \cap \phi_j} Q(s_z, a_z = 0)$ on both sides.

The detailed proof is provided in Appendix B. $\qquad \square$

Thus when $k = 1$, selecting the top arm according to the $\lambda$ value is equivalent to maximizing the cumulative long-term reward, and is guaranteed to be optimal.

**Theorem 2** *When $k > 1$, selecting the $k$ arms is a NP-hard problem. The non-asymptotic tight upper bound and non-asymptotic tight lower bound for getting the optimal solution are $o(C(n, k))$ and $\omega(N)$, respectively.*

*Proof Sketch.* This problem can be considered as a variant of the knapsack problem. If we disregard the influence of the shared neighbor nodes for two selected arms, then selecting arm $i$ will not influence the future selection of arm $j$. In such instances, the problem of selecting the $k$ arms is simplified to the traditional 0/1 knapsack problem, a classic NP-hard problem. Therefore, when considering the effect of shared neighbor nodes for two selected arms, this problem is at least as challenging as the 0/1 knapsack problem. $\qquad \square$

When $k > 1$, it is difficult to compute the optimal solution, we provide a heuristic greedy algorithm with the complexity of $O(\frac{(2N-k)*k}{2})$ in Section C in the appendix.

## 5 Experiment

In this section, we demonstrate the effectiveness of EduQate against benchmark algorithms on synthetic students and students derived from a real-world dataset, the Junyi Dataset and the OLI Statics dataset. All experiments are run on CPU only. In our experiments, we compare EduQate with the following policies:

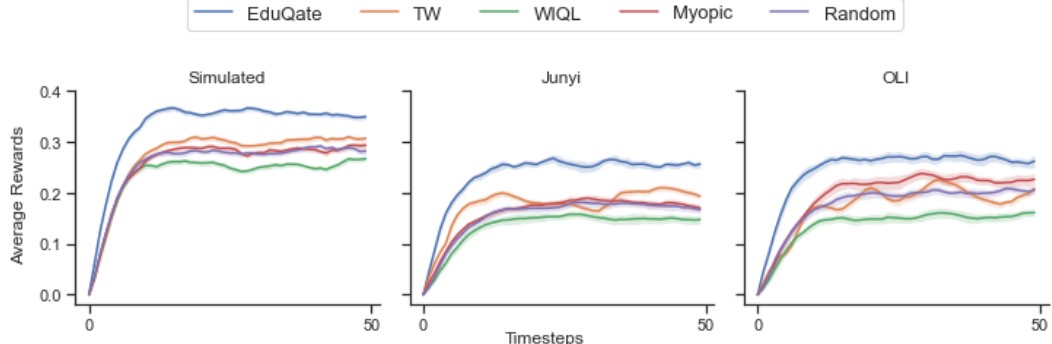

Figure 1: Average rewards for the respective algorithms on 3 datasets, averaged across 30 runs. Shaded regions represent standard error.

- **Threshold Whittle (TW)**: This algorithm, proposed by [17], utilizes an efficient closed-form approach to compute the Whittle index, considering only the pull action as active. It operates under the assumption that transition probabilities are known and stands as the state-of-the-art in RMABs.
- **WIQL**: This algorithm employs a Q-learning-based Whittle Index approach [5]. It learns Q-values using the pull action as the only active strategy and calculates the Whittle Index based on the acquired Q-values.
- **Myopic**: This strategy disregards the impact of the current action on future rewards, concentrating solely on predicted immediate rewards. It selects the arm that maximizes the expected reward at the immediate time step.
- **Random**: This strategy randomly selects arms with uniform probability, irrespective of the underlying state.

Inspired by work in healthcare settings [12, 14], we compare the policies by the *Intervention Benefit* (*IB*), as shown in the following equation:

$$IB_{Random,EQ}(\pi) = \frac{\mathbb{E}_{\pi}(R(.)) - \mathbb{E}_{Random}(R(.))}{\mathbb{E}_{EQ}(R(.)) - \mathbb{E}_{Random}(R(.))} \tag{4}$$

where *EQ* represents EduQate, and *Random* represents a policy where the arms are selected at random. Prior work in educational settings has demonstrated that random policies can yield robust learning outcomes through spaced repetition [9, 10]. Therefore, to establish efficacy, successful algorithms must demonstrate superiority over random policies. Our chosen metric, *IB*, effectively compares the extent to which a challenger algorithm $\pi$ outperforms a random policy in comparison to our algorithm.

## 5.1 Experiment setup

In all experiments, we commence by initializing all arms in state 0 and permit the teacher algorithms to engage with the student for a total of 50 actions, pulling exactly 1 arm (i.e. $k = 1$) at each time step. Following the completion of these actions, the episode concludes, and the student state is reset. This process is iterated across 800 episodes, for a total of 30 seeds. The datasets used in our experiment are described below:

**Synthetic dataset.** Given the domain-motivated constraints on the transition functions highlighted in Section 3.1, we create a simulator based on $N = 50$, $S \in \{0, 1\}$, $N_{topics} = 20$. We randomly assign arms to topic groups, and allow arms to be assigned to be more than one topic. Under this method, number of arms under each group may not be equal. For each trial, a new transition matrix is generated to simulate distinct student scenarios.

**Junyi dataset.** The Junyi dataset [7] is an extensive dataset collected from the Junyi Academy [1], an eLearning platform established in 2012 on the basis of the open-source code released by Khan

---
[1]http://www.Junyiacademy.org/

Table 1: Comparison of policies on synthetic, Junyi, and OLI datasets. $\mathbb{E}[R]$ represents the average reward obtained in the final episode of training. Statistic after $\pm$ represents standard error across 30 trials.

| Policy | Synthetic | | Junyi | | OLI | |
|---|---|---|---|---|---|---|
| | $\mathbb{E}[IB](\%)\pm$ | $\mathbb{E}[R]\pm$ | $\mathbb{E}[IB](\%)\pm$ | $\mathbb{E}[R]\pm$ | $\mathbb{E}[IB](\%)\pm$ | $\mathbb{E}[R]\pm$ |
| Random | - | $26.84 \pm 0.46$ | - | $15.82 \pm 0.34$ | - | $18.46 \pm 0.35$ |
| WIQL | $-49.03 \pm 15.07$ | $24.60 \pm 0.43$ | $-26.77 \pm 7.39$ | $14.01 \pm 0.97$ | $-60.20 \pm 19.38$ | $14.33 \pm 0.42$ |
| Myopic | $-3.44 \pm 5.81$ | $27.07 \pm 0.52$ | $10.74 \pm 3.13$ | $16.86 \pm 0.356$ | $39.92 \pm 12.00$ | $20.51 \pm 0.48$ |
| TW | $37.21 \pm 17.02$ | $28.50 \pm 0.47$ | $31.284 \pm 2.65$ | $15.819 \pm 0.34$ | $0.20 \pm 9.27$ | $18.07 \pm 0.21$ |
| **EduQate** | **100.0** | **$34.33 \pm 0.49$** | **100.0** | **$24.53 \pm 0.31$** | **100.0** | **$25.47 \pm 0.47$** |

Academy. In this dataset, there are nearly 26 million student-exercise interactions across 250 000 students in its mathematics curriculum. For this experiment, we selected the top 100 exercises with the most student interactions to create our student models. Using our method to generate groups, the resultant EdNetRMAB has $N = 100$ and $N_{topics} = 21$.

**OLI Statics dataset.** The OLI Statics dataset [4] comprises student interactions with an online Engineering Statics course[2]. In this dataset, each item is assigned one or more Knowledge Components (KCs) based on the related topics. After filtering for the top 100 items with the most student interactions, the resultant EdNetRMAB includes $N = 100$ items and $N_{topics} = 76$ distinct topics.

## 5.2 Creating student models

In this section, we outline the procedure for generating student models aimed at simulating the learning process. To clarify, a student model in this context is defined as a set of transition matrices for all items. These matrices are employed with EdNetRMABs to simulate the learning dynamics.

We employ various strategies to model transitions within the RMAB framework. Active transitions are determined by assessing the average success rate on a question before and after a learning intervention. Passive transitions are influenced by difficulty ratings, with more challenging questions more prone to rapid forgetting. Semi-active transitions, on the other hand, are computed as proportion of active transition, guided by similarity scores. Here, we provide an outline and the full details can be found in Appendix D.

**Active Transitions.** We use data on students' correct response rate after interacting with an item to create the transition matrix for action 2, based on the change in correctness rates before and after a learning intervention.

**Passive Transitions.** To construct passive transitions for items, we use relative difficulty scores to determine transitions based on difficulty levels. We assume that higher difficulty correlates with a greater likelihood of forgetting, resulting in higher failure rates. Specifically, higher difficulty values correspond to higher $P_{1,0}^0$ values, indicating a greater likelihood of forgetting. The transition matrix for the passive action $a = 0$ is then randomly generated, with values influenced by difficulty levels.

**Semi-active Transitions.** To derive semi-active transitions, we use similarity scores between exercises from the Junyi dataset. We first normalize these scores to the range $[0, 1]$. Then, for any chosen arm, we compute its transition matrix under the semi-active action $a = 1$ as a proportion of its active action transitions, $P_{0,1}^1 = \sigma(P_{0,1}^2)$, where $\sigma$ signifies the similarity proportion.

The arm's transition matrix for the semi-active action varies due to different similarity scores between pairs in the same group. To address this, we use the average similarity score to determine the proportion. Since the OLI dataset does not contain similarity ratings, we assume a constant similarity rating of $\sigma = 0.8$ for all pairs.

## 6 Results

The experimental results for the synthetic, Junyi, and OLI datasets are shown in Table 1. We report the average intervention benefit $IB$ and final episode rewards from thirty independent runs for five algorithms: EduQate, TW, WIQL, Myopic, and Random. EduQate consistently outperforms the other policies across all datasets, demonstrating higher intervention benefits and average rewards.

---

[2]https://oli.cmu.edu/courses/engineering-statics-open-free/

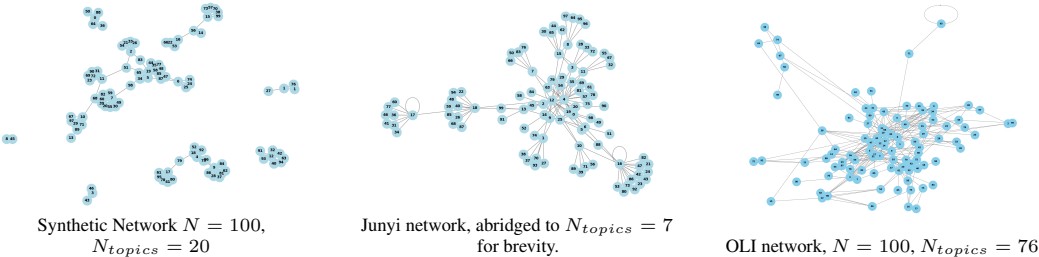

Synthetic Network $N = 100$,      Junyi network, abridged to $N_{topics} = 7$     
$N_{topics} = 20$      for brevity.      OLI network, $N = 100$, $N_{topics} = 76$.

Figure 2: This visualization compares network complexities from our experiments. The synthetic dataset (left) shows simpler, isolated groups, while the real-world datasets (Junyi, middle; OLI,right) displays more intricate and interconnected relationships amongst items.

In terms of $IB$, we note that all challenger policies do not exceed 50%, indicating two key points. First, as noted in prior works [9], our results confirm that random policies in educational settings are robust and difficult to surpass, even when algorithms are equipped with knowledge of the learning dynamics. Second, our interdependency-aware EduQate performs well over random policies and other algorithms, highlighting the importance of considering network effects and interdependencies in EdNetRMABs.

Notably, WIQL, which relies solely on Q-learning for active and passive actions, performs worse than a random policy, likely due to misattributing positive network effects to passive actions. Despite having access to the transition matrix, TW does not perform as well as the interdependency-aware EduQate. While it has demonstrated effectiveness in traditional RMABs, TW weaknesses become evident in the current setting, where pulling an arm has wider implications to other arms. Overall, EduQate has demonstrated robust and effective performance in maximizing rewards across different datasets. Figure 1 shows the average rewards obtained in the final episode for each algorithm.

Figure 2 provides visualizations of the networks generated from synthetic students and mined from real-world datasets. The synthetic dataset produces networks with distinct isolated groups, contrasting with the more intricate and interconnected networks from the Junyi and OLI datasets, reflecting real-world complexities. Despite these differing topologies and levels of interdependency, EduQate performs well under all network setups. In Appendix E.1, we explore the effects of different network topologies by varying the number of topics while limiting the membership of each item. We find that as network interdependencies are reduced, the network effects diminish, and such EdNetRMABs can be approximated to traditional RMABs with independent arms. Under these conditions, our algorithm does not perform as well as other baseline policies.

Finally, an ablation study detailed in Appendix E.2 examines the effectiveness of the replay buffer in EduQate. The study shows that the replay buffer helps overcome the cold-start problem, where initial learning episodes provide sub-optimal experiences for students [3].

## 7 Conclusion and Limitations

In this paper, we introduced EdNetRMABs to the education setting, a variant of MAB designed to model interdependencies in educational content. We also proposed EduQate, a novel Whittle-based learning algorithm tailored for EdNetRMABs. Unlike other Whittle-based algorithms, EduQate computes an optimal policy without requiring knowledge of the transition matrix, while still accounting for the network effects of pulling an arm. We demonstrated the guaranteed optimality of a policy trained under EduQate and showcased its effectiveness on synthetic and real-world datasets, each with its own characteristic.

Our work assumes that student knowledge states are fully observable and available at all times, which is a limitation. Despite this, we believe our work is significant and can inspire further research to improve efficiencies in education. For future work, we aim to extend EduQate to handle partially observable states and address the cold-start problem in education systems by minimizing the initial exploratory phase.

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

 **Appendix/Supplementary Materials**

 **A    Table of Notations**

Table 2: Notations

| Notation | Description |
|---|---|
| $N, N_{topics}$ | $N$: number of arms in EdNetRMABs; $N_{topics}$: number of topic groups |
| $s_i^t$ | $s_i^t$: state of arm $i$ at time step $t$. 1: learned, 0: unlearned. |
| $a_i^t$ | $a_i^t$: action of arm $i$ at time step $t$. 0: passive action, 1: semi-active action, 2: active action. |
| $\mathbf{s}, \mathbf{a}$ | $\mathbf{s}, \mathbf{a}$: joint state vector and joint action vector of EdNetRMABs. |
| $\phi_i, \phi_i^-$ | $\phi_i$: the set of arms that includes the arm $i$ and its connected neighbors, $\phi_i^-$: $\phi_i$ that exclude arm $i$. |
| $P_{s,s'}^{i,a}$ | $P_{s,s'}^{i,a}$ is the probability of transition from state $s$ to $s'$ when arm $i$ is taking action $a$. |
| $Q_i(s_i, a_i)$ | $Q_i(s_i, a_i)$ is the state-action value function for the arm $i$ when taking action $a_i$ with state $s_i$. |
| $V_i(s_i)$ | The value function for arm $i$ at the state $s_i$. |

 **B    Proof for the theorem**

 We rewrite the theorem here for ease of explanation.

 **Theorem 3** *Choose top arms according to the $\lambda$ value in Equation 1 is equivalent to maximize the*
 *cumulative long-term reward.*

 *Proof.*    According to the approach, we select the arm according to the $\lambda$ value. Assume arm $i$ has
 the highest $\lambda$ value, then for any arm $j$, where $i \neq j$, we have

$$\lambda_i \geq \lambda_j$$

$$Q(s_i, a_i = 1) - Q(s_i, a_i = 0) + \sum_{i \in \phi_i^-} (Q(s_i, a_i = 1) - Q(s_i, a_i = 0)) \geq Q(s_j, a_j = 1) - Q(s_j, a_j = 0) + \sum_{j \in \phi_j^-} (Q(s_j, a_j = 1) - Q(s_j, a_j = 0))$$

$$Q(s_i, a_i = 1) + \sum_{i \in \phi_i^-} (Q(s_i, a_i = 1)) + Q(s_j, a_j = 0) + \sum_{j \in \phi_j^-} (Q(s_j, a_j = 0)) \geq Q(s_j, a_j = 1) + \sum_{j \in \phi_j^-} (Q(s_j, a_j = 1)) + Q(s_i, a_i = 0) + \sum_{i \in \phi_i^-} (Q(s_i, a_i = 0))$$

$$(5)$$

 There are two cases:

 • **arm $i$ and arm $j$ are not connected, and group $\phi_i$ and $\phi_j$ has no overlap, i.e.,** $\phi_i \cap \phi_j = \emptyset$. We
 add $\sum_{z \notin \phi_i \wedge z \notin \phi_j} Q(s_z, a_z = 0)$ on both sides, we can have the left side:

$$Q(s_i, a_i = 1) + \sum_{i \in \phi_i^-} (Q(s_i, a_i = 1)) + Q(s_j, a_j = 0) + \sum_{j \in \phi_j^-} (Q(s_j, a_j = 0)) + \sum_{z \notin \phi_i \wedge z \notin \phi_j} Q(s_z, a_z = 0)$$

$$= Q(s_i, a_i = 1) + \sum_{i \in \phi_i^-} (Q(s_i, a_i = 1)) + \sum_{j \notin \phi_i^-} (Q(s_j, a_j = 0))$$

$$= Q(\mathbf{s}, \mathbf{a} = \mathbb{I}_i)$$

$$(6)$$

 Similarly, the right side becomes

$$Q(s_j, a_j = 1) + \sum_{j \in \phi_j^-} (Q(s_j, a_j = 1)) + \sum_{i \notin \phi_j} (Q(s_i, a_i = 0)) = Q(\mathbf{s}, \mathbf{a} = \mathbb{I}_j) \qquad (7)$$

 Thus, the equation 2 becomes

$$Q(\mathbf{s}, \mathbf{a} = \mathbb{I}_i) \geq Q(\mathbf{s}, \mathbf{a} = \mathbb{I}_j) \qquad (8)$$

 • **arm $i$ and arm $j$ are not connected, but group $\phi_i$ and $\phi_j$ has overlap, i.e.,** $\phi_i \cap \phi_j \neq \emptyset$. In this
 case, we add $\sum_{z \notin \phi_i \wedge z \notin \phi_j} Q(s_z, a_z = 0) - \sum_{z \in \phi_i \cap \phi_j} Q(s_z, a_z = 0)$ on both sides, we can have the

left side:

$$Q(s_i, a_i = 1) + \sum_{i \in \phi_i^-} (Q(s_i, a_i = 1)) + Q(s_j, a_j = 0) + \sum_{j \in \phi_j^-} (Q(s_j, a_j = 0)) + \sum_{z \notin \phi_i \wedge z \notin \phi_j} Q(s_z, a_z = 0) - \sum_{z \in \phi_i \cap \phi_j} Q(s_z, a_z = 0)$$

$$= Q(s_i, a_i = 1) + \sum_{i \in \phi_i^-} (Q(s_i, a_i = 1)) + \sum_{j \in \phi_j} (Q(s_j, a_j = 0)) + \sum_{z \notin \phi_i \wedge z \notin \phi_j} Q(s_z, a_z = 0) - \sum_{z \in \phi_i \cap \phi_j} Q(s_z, a_z = 0)$$

$$= Q(s_i, a_i = 1) + \sum_{i \in \phi_i^-} (Q(s_i, a_i = 1)) + \sum_{j \notin \phi_i^-} (Q(s_j, a_j = 0))$$

$$= Q(\mathbf{s}, \mathbf{a} = \mathbb{I}_i)$$

$$(9)$$

Similarly, the right side becomes

$$Q(s_j, a_j = 1) + \sum_{j \in \phi_j^-} (Q(s_j, a_j = 1)) + \sum_{i \notin \phi_j} (Q(s_i, a_i = 0)) = Q(\mathbf{s}, \mathbf{a} = \mathbb{I}_j) \qquad (10)$$

- **arm $i$ and arm $j$ are connected, and group $\phi_i$ and $\phi_j$ has overlap, i.e., $\phi_i \cap \phi_j \neq \emptyset$, and** $\{i, j\} \subset \phi_i \cap \phi_j$. This case is similar to the previous one, we add $\sum_{z \notin \phi_i \wedge z \notin \phi_j} Q(s_z, a_z = 0) -$ $\sum_{z \in \phi_i \cap \phi_j} Q(s_z, a_z = 0)$ on both sides, we can have the left side: $Q(\mathbf{s}, \mathbf{a} = \mathbb{I}_i)$ and the right side $Q(\mathbf{s}, \mathbf{a} = \mathbb{I}_j)$.

$\square$

We show that, using Theorem 1, selecting the top arms according to the $\lambda$ value is guaranteed to maximize the cumulative long-term reward, thus proving it to be optimal.

However when it comes to the case where $k > 1$, selecting the top $k$ arms according to the $\lambda$ value is not guaranteed to be optimal. Let the $\Phi$ denote the set of arms that are selected, i.e., $a_i = 2$ if $i \in \Phi$. Because once the arm $i$ is added to the selected arm set $\Phi$, the benefit of selecting arm $j$ will also be influenced if the arm $j$ has the shared connected neighbor arms with arm $i$, i.e., $\phi_i \cap \phi_j \neq \emptyset$. To this end, finding the optimal solution is difficult, as we need to list all the possible solution sets. The non-asymptotic tight upper bound and non-asymptotic tight lower bound for getting the optimal solution are $o(C(n, k))$ and $\omega(N)$, respectively.

We provide the proof for Theorem 2: *Proof.* When considering the influence of the shared neighbor nodes for two selected arms, then selecting arm $i$ will influence the future benefit of selecting arm $j$ if arm $i$ and arm $j$ have the overlapped neighbor nodes, i.e., $\phi_i \cap \phi_j \neq \emptyset$. This is because the calculation of $\lambda_j$, as some arms $z \in \phi_i \cap \phi_j$ already receive the semi-active action $a = 1$ due to the selection of arm $i$, the subsequent selection of arm $j$ would not double introduce the benefit from those arms $z$ who already included in $\phi_i$. However, if the top $k$ arms ranked according to their $\lambda$ value do not have any overlaps in their connected neighbor nodes, i.e, $\phi_i \cap \phi_j = \emptyset$ for $\forall i, j$, where arm $i$ and arm $j$ are top $k$ arms according to $\lambda$ value. We can directly add those top $k$ arms to the action set $\Phi$, and the solution is guaranteed to be optimal. Then we have the non-asymptotic tight lower bound for getting the optimal solution which is $\omega(N)$. Otherwise, if the top $k$ arms ranked according to their $\lambda$ value have any overlaps in their connected neighbor nodes, to get the optimal solutions, we need to list all possible combinations of the $k$ arms, which have the $C(n, k)$ cases, and computing the corresponding sum of the $\lambda$ value. In this case, we can derive that the non-asymptotic tight upper bound for getting the optimal solution is $o(C(n, k))$. $\square$

## C   Greedy algorithm when $k > 1$

When $k > 1$, it is difficult to compute the optimal solution as we might list all possible solutions, and the complexity is $O(C(n, k))$, Thus we provide a heuristic greedy algorithm to find the near-optimal solutions. The process to decide the selected arm set $\Phi$ is as follows:

1. We first compute the independent $\lambda$ value for each arm $i$, where $i \in \{1, \ldots, N\}$, where $\lambda_i = Q(s_i, a_i = 1) - Q(s_i, a_i = 0) + \sum_{j \in \phi_i^-} (Q(s_i, a_i = 2) - Q(s_i, a_i = 0))$;

2. We add the arm with the top $\lambda$ value to the set $\Phi$;

3. We recompute the $\lambda$ value for the each arm, note that we will remove $Q(s_j, a_j)$ in the $\lambda$ equation if $j \in \Phi$ or $j \in \phi_j$ for $\forall i \in \Phi$;

4. we add the arm with the top $\lambda$ value to the set $\Phi$, and repeat the step 3 and 4 until we add $k$ arms to set $\Phi$.

The intuition of such a heuristic greedy algorithm is to add the arm that maximizes the marginal gain to the action. And the complexity for the greedy algorithm is $O(\frac{(2N-k)*k}{2})$.

# D   Generating Student Models from Junyi and OLI Dataset

In this section, we describe the features in Junyi and OLI dataset which we use in developing the transition matrices.

The datasets contain the following features which we use in various aspects to generate the student models and the network:

- Topic & Knowledge Component Classification: Items are classified into topics (Junyi) or KCs (OLI). This classification is employed to group items and establish the initial network.

- Similarity: The Junyi dataset offers expert ratings for exercise similarity, enabling a nuanced approach to form richer group memberships. High similarity scores group exercises together, irrespective of topic tags.

- Difficulty: The Junyi dataset provides expert ratings to determine the relative difficulty of exercise pairs. In the OLI dataset, we use the overall correct response rate as a measure of difficulty.

- Rate of Correctness: By analyzing student-exercise interactions, we calculate the frequency of correct answers for each question, offering insights into the improvement of knowledge over time.

## D.1   Active Transitions

**Junyi Dataset**   The Junyi dataset contains `earned_proficiency` feature which indicates if the student has achieved mastery of the topic based on Khan Academy's algorithm[3]. Thus, we take the number of attempts before `earned_proficiency=1` as $P^2_{0,1}$, and the errors made during mastery as $P^2_{1,0}$.

**OLI Dataset**   We possess records of students' accuracy on quiz questions after studying specific topics. To derive the transition matrix for the student with the corresponding action 2, we utilize the change in correctness rate before and after a learning intervention.

Given that proportion of correct attempts at time $t$ as $a^t$, then $a^{t+1} = P^2_{0,1}(1 - a^t) + P^2_{1,1}(a^t)$. We use a linear regressor to estimate the respective $P^2$, constraining it to produce positive values and clipping the values to $0.99$ when required.

## D.2   Passive Transitions

To construct passive transitions for exercises, we utilize relative difficulty scores to determine transitions based on difficulty levels. We operate under the assumption that the difficulty of an exercise is linked to its likelihood of being forgotten, thereby resulting in a higher failure rate. More precisely, higher difficulty values of an exercise correspond to higher $P^0_{1,0}$ values, indicating a greater likelihood of forgetting. The transition matrix for the passive action $a = 0$ is then randomly generated, with the values influenced by the difficulty levels.

## D.3   Semi-active Transitions

To derive semi-active transitions, the Junyi dataset contains similarity scores between two distinct exercises, quantifying their similarity on a 9-point Likert scale. Once the transition matrices are computed under the active action $a = 2$ for all arms, we proceed to calculate the transition matrix

---

[3]http://david-hu.com/2011/11/02/how-khan-academy-is-using-machine-learning-to-assess-student-mastery.html

for the semi-active action $a = 1$. This involves normalizing the similarity scores to the range $[0, 1]$, denoted as $\sigma$. For any chosen arm/topic, we can then compute its neighbor's transition matrix under the semi-active action $a = 1$ with $P_{0,1}^1 = \sigma(P_{0,1}^2)$, where $\sigma$ signifies the similarity proportion. It is worth noting that an arm's transition matrix for the semi-active action varies due to different neighbors being selected — different neighbors correspond to different similarity scores.

To address this, we can store the transition matrix of semi-active actions for different neighbor selection scenarios, preserving the flexibility of our algorithm. In this work, for simplicity, we opt not to distinguish the impact of different neighbors being selected. Instead, we calculate the average similarity for all arms in a group average them, and use the resultant average as $\sigma$.

For the OLI Statics dataset, we use a constant value of $\sigma = 0.8$ since there are no similarity scores available.

## E    Additional Experiment Results and Discussion

### E.1    Comparing Different Network Setups

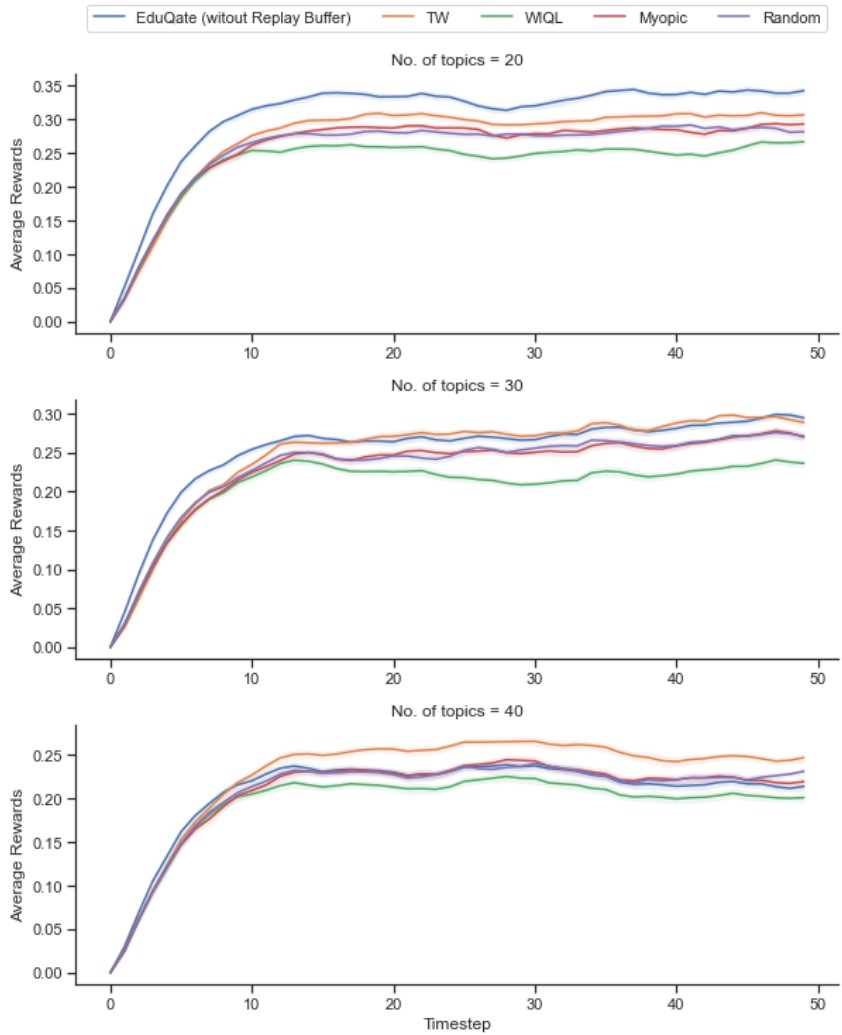

Figure 3: Average rewards for the respective algorithms, on the last episode of training. Note that as $N_{topics}$ increase, the network effects are reduced, and most algorithms are not better than a random policy.

Table 3: Comparison of policies on synthetic dataset, with different network setups. Note that that as $N_{topics}$ increase, the reliability of any algorithms decreases, as seen by the standard deviations of their average $IB$. EduQate- here refers to the EduQate algorithm without replay buffer.

| $N_{topics}$ | POLICY | $\mathbb{E}[IB]$ (%) ($\pm$) |
|---|---|---|
| 20 | WIQL | $-57.9 \pm 13.1$ |
| | MYOPIC | $0.24 \pm 8.2$ |
| | TW | $32.6 \pm 7.0$ |
| | EDUQATE- | **100.0** |
| 30 | WIQL | $-292 \pm 1162$ |
| | MYOPIC | $180 \pm 600$ |
| | TW | $122 \pm 277$ |
| | EDUQATE- | $100$ |
| 40 | WIQL | $307 \pm 1069$ |
| | MYOPIC | $212 \pm 526$ |
| | TW | $4.34 \pm 1124$ |
| | EDUQATE- | $100$ |

We present the results for different network setups in Table 3. We note that as the number of topics approach the number of arms (i.e. $N_{topics} = \{30, 40\}$, all algorithms perform in a highly unstable manner, as reflected in the standard deviations presented. We emphasizes here that the performance of EduQate is dependent on the quality of the network it is working on, and tends to thrive in more complex, yet realistic scenarios, such as the Junyi dataset presented in Figure 2. We present an example of a graph generated when $N_{topics} = 40$ in Figure 4, where we notice that many arms do not belong to a group. Under this network, the EdNetRMAB can be approximated to a traditional RMAB, where the arms are independent of each other.

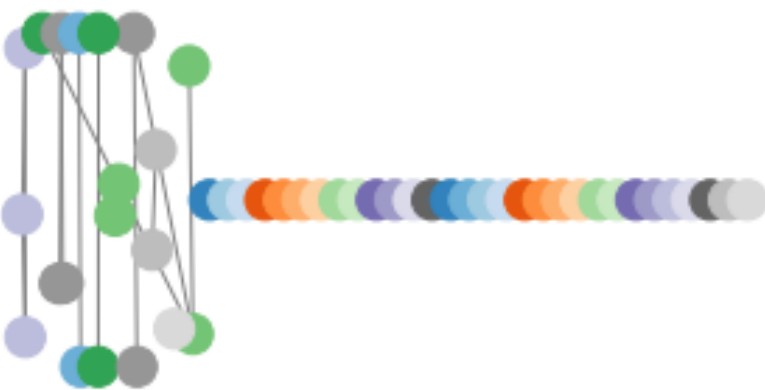

Figure 4: Synthetic network when $N_{topics} = 40$. Note that some arms are without group members, and do not receive benefits from networks. Node colors represent topic groups.

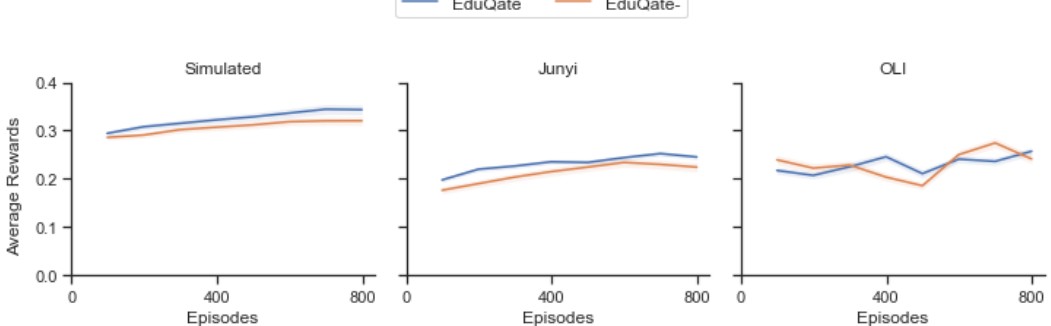

Figure 5: Average rewards across 800 episodes of training, across 30 seeds. EduQate- (orange) refers to the EduQate algorithm without replay buffer.

### E.2 Ablation of Replay Buffer

Table 4: Comparison of EduQate with and without (EduQate-) Experience Replay Buffer policies across different datasets. Results reported are of the final episode of training.

| POLICY | $\mathbb{E}[IB]$ (%) $\pm$ | | |
|---|---|---|---|
| | SYNTHETIC | JUNYI | OLI |
| EDUQATE- | $104.74 \pm 32.56$ | $76.90 \pm 4.72$ | $107.30 \pm 11.77$ |
| EDUQATE | 100.0 | 100.0 | 100.0 |
| POLICY | $\mathbb{E}[R] \pm$ | | |
| | SYNTHETIC | JUNYI | OLI |
| EDUQATE- | $32.032 \pm 0.469$ | $22.133 \pm 0.544$ | $25.16 \pm 0.432$ |
| EDUQATE | $34.331 \pm 0.489$ | $24.527 \pm 0.314$ | $25.468 \pm 0.469$ |

We investigate the importance of the Experience Replay buffer in EduQate, as shown in Figure 5 and Table 4. For the Simulated and Junyi datasets, EduQate without Experience Replay (EduQate-) does not achieve the performance levels of the full EduQate algorithm within 800 episodes, highlighting the importance of methods that aid Q-learning convergence. In real-world applications, slow convergence can result in students experiencing a curriculum similar to a random policy, leading to sub-optimal learning experiences during the early stages. This issue is known as the cold-start problem [3]. Future work in EdNetRMABs should explore methods to overcome cold-start problems and improve convergence in Q-learning-based methods.

## F    Q-Learning

Q-learning [25] is a popular reinforcement learning method that enables an agent to learn optimal actions in an environment by iteratively updating its estimate of state-action value, $Q(s, a)$, based on the rewards it receives. The objective, therefore, to learn $Q^*(s, a)$ for each state-action pair of an MDP, given by:

$$Q^*(s, a) = r(s) + \sum_{s' \in S} P(s, a, s') \cdot V^*(s')$$

where $V^*(s')$ is the optimal expected value of a state, is given by:

$$V^*(s) = max_{a \in A}(Q(s, a))$$

Q-learning estimates $Q^*$ through repeated interactions with the environment. At each time step $t$, the agent takes an action $a$ using its current estimate of $Q$ values and current state $s$, thus received a reward of $r(s)$ and new state $s'$. Q-learning then updates the current estimate using the following:

$$\begin{aligned}
Q_{new}(s,a) \leftarrow &(1-\alpha) \cdot Q_{old}(s,a) \\
&+ \alpha \cdot (r(s) \\
&+ \gamma \cdot max_{a \in A} Q_{old}(s',a))
\end{aligned} \tag{11}$$

where $\alpha \in [0,1]$ is the learning rate that controls updates, and $\gamma$ is the discount on future rewards associated with the MDP.

# G   Experiment Details and Hyperparameters

| Category | Parameter | Value |
|---|---|---|
| Replay buffer | buffer_size | 10000 |
| | batch_size | 64 |
| WIQL/EduQate | $\gamma$ | 0.95 |
| | $\alpha$ | 0.1 |

Table 5: Hyperparameters for Replay Buffer and Q-learning

