# OpenReview forum: "EduQate: Generating Adaptive Curricula through RMABs in Education Settings"
_NeurIPS.cc/2024/Conference — Submitted to NeurIPS 2024_

### Official Review · Reviewer_MTkZ · 2024-07-04

**Soundness:** 2
**Presentation:** 3
**Contribution:** 2
**Rating:** 3
**Confidence:** 3

**Summary:**

This paper proposes a variant of Multi-Armed Bandits (MAB) named EdNetRMABs that model the interdependencies between learning contents to meet the real-world education scenarios. Subsequently, the authors introduce EduQate, an interdependency-aware Q-learning algorithm to optimize content recommendation given the EdNetRMABs. The paper demonstrates the theoretical and empirical effectiveness of EduQate with the experimental results of synthetic and real-world data.

**Strengths:**

1. This work uses RMABs in education to model the learning process with interdependent educational content as the different knowledge concepts have relevance.
2. The paper introduces EduQate employs Q-learning to make decisions on arm selection that do not require knowledge of the transition matrix to compute an optimal policy and provides related theoretical analysis.

**Weaknesses:**

1. As this work considers interdependency awareness in content recommendation in educational scenarios, the generated group in the experiment is not clear. The authors need to clarify how to capture relevance between the selected exercises of each topic in different datasets. Furthermore, does the different number of topics influence the experimental results?
2. The definition of state space representing a student's knowledge states a binary value. However, in real-world scenarios, the knowledge states are multi-level. A fine-grained knowledge state estimation result is essential for the content recommendation in adaptive learning.
3. The paper lacks the details of EdNetRMABs. Some illustrations would help to understand.

**Questions:**

NA

---

> ### Author Rebuttal · Authors · 2024-08-06
>
> We regret that Reviewer MTkZ did not find our work to be a meaningful contribution. We hope that our responses have addressed the reviewer's concerns. We welcome the opportunity for further discussion to clarify any doubts and provide additional insights into our work. Additionally, we would like to address the weaknesses mentioned by the reviewer.
>
> ## Weaknesses
> 1. On generating groupings in experiments:
> The Junyi and OLI datasets categorize each exercise in their question banks into topics, which we exploit to group the exercises. Additionally, Junyi provides expert ratings of similarities between questions, which we use to further augment the groupings. Further details on deriving the student models can be found in Appendix D (lines 474-489). We agree that this important detail should be clarified in the main text, and we will include it in future iterations.
> Regarding the number of topics, we provide an ablation study in Appendix E, which examines the simulated and highly constrained situation where $N_{topics}$ = {30,40}. We show that as $N_{topics}$ approaches N, the EdNetRMAB can be approximated to a traditional RMAB, where arms are independent of each other (Figure 4). In such cases, all algorithms perform in a highly unstable manner, as seen in Table 3 of the original paper. In contrast, under realistic scenarios generated from the real-world datasets Junyi and OLI, EduQate outperforms all other algorithms (Table 1 of the main text).
>
> 2. On binary state spaces:
> We agree with the reviewer that having a binary state space is a limiting factor of our work. Even though knowledge states can be complex and continuous, these are not observable and requires many assumptions to model. The observations are only at the level of answers to questions posed and these are binary. The complex knowledge states would have to be learnt based on the binary observations received over time. While we consider only single step observation here, in the future, we can consider multi-step observations to reason about this partial observability (of the knowledge state). We wish to point out that even with single step observation where we consider future impact and interrelationships, we are able to outperform existing methods.
>
> 3. On EdNetRMABs:
> We provide an illustration of EdNetRMABs in the attached PDF, Figure 1. Additionally, we invite the reviewer to refer to Section 3.1 for full details on EdNetRMABs, which we consider one of the core contributions of this work. We have made every effort to detail it thoroughly. We are happy to address any confusion and provide additional clarifications if needed.

---

### Official Review · Reviewer_fqUb · 2024-07-12

**Soundness:** 3
**Presentation:** 3
**Contribution:** 2
**Rating:** 4
**Confidence:** 4

**Summary:**

The paper introduces **EduQate**, a system that generates adaptive educational curricula using restless multi-armed bandits (RMABs). This method aims to efficiently achieve mastery across multiple interdependent educational contents. Unlike traditional methods that assume learning contents are independent, EduQate acknowledges and leverages the interdependencies between different educational concepts.

The paper addresses key challenges in modeling and optimizing the learning process in educational environments where the learning of different topics is interconnected. By considering these interdependencies, the proposed EduQate system ensures more accurate and effective personalized learning.

**Strengths:**

**Originality** The paper introduces the concept of EdNetRMABs, a model that considers the interdependencies among learning contents in educational settings. This approach contrasts with existing methods that typically assume independence among different educational topics. By employing the Whittle index and Q-learning to develop the EduQate algorithm, the paper further demonstrates originality in adapting and integrating these techniques to address the specific challenges of personalized education.

**Quality** The authors provide clear definitions and a well-structured EdNetRMABs model. The optimality guarantees of the EduQate algorithm are well-explained and supported through rigorous proofs. Additionally, the empirical results are compelling, demonstrating the effectiveness of the proposed method over benchmark strategies using both synthetic and real data.

**Clarity** The paper is clearly written and well-organized, making complex concepts easier to understand. The authors systematically introduce the problem, the proposed solution, and both theoretical and empirical validations. The use of figures helps to elucidate the model and results. However, the clarity could be further enhanced by including more detailed explanations of the algorithm's implementation and practical applications.

**Significance** The significance of the paper lies in its potential impact on the field of educational technology. By addressing the interdependencies among learning contents, EduQate offers a more accurate and efficient approach to personalized education.

**Weaknesses:**

1. **Clarity of Explanations**: Due to the limited length, many explanations in the main text are not very clear. For instance, the survival and design of the student model and the content and information of the datasets are not thoroughly explained. For example, there is a dataset mentioned later that lacks similarity content, which should have been noted when introducing the dataset.

2. **Inconsistency in Notation**: In Section 4.1, "Analysis of EduQate," the variable k in the second line has a different font from the k mentioned later. Additionally, the content related to k mentioned earlier is too far from this section, making it difficult for readers to understand and potentially leading to misunderstandings.

3. **Simplified Modeling**: The model uses only one pseudo-state, where if any content within a group is learned, all unlearned content in the group is marked as pseudo-state. This approach seems overly simplified. For example, in Case 1, if only one piece of content in a group is learned, all other content is marked as pseudo-state. In Case 2, if many pieces of content in a group are learned, the remaining content is also marked as pseudo-state. Although both cases result in a pseudo-state, their actual significance differs, and it seems the authors did not consider this distinction.

4. **Bidirectional Relationships**: The relationships between knowledge points in the paper are bidirectional, with only grouping relationships. However, in practice, many relationships between knowledge points within the same group are unidirectional. Recommending subsequent courses without recommending prerequisite courses first can lead to students needing to self-study prerequisite courses when learning subsequent courses, increasing their learning burden. The modeling in the paper does not seem to account for such unidirectional structures, instead using groups to link knowledge points, which could lead to this reverse learning issue.

5. **User Experience**: The discussion on user experience for educators and students is insufficient. Including qualitative and quantitative feedback from pilot implementations would be useful. The authors should conduct user research to gather insights on the intuitiveness and user-friendliness of the system for the target users. This could include surveys, interviews, and usability testing, focusing on ease of use, effectiveness, and areas needing improvement.

6. **Scalability of the EduQate Algorithm**: Although the paper mentions the efficiency of the EduQate algorithm, it lacks an in-depth exploration of its scalability in large-scale educational environments. Detailed performance benchmarking of the algorithm, including analysis of computation time and resource utilization, would be beneficial. The authors should provide a comprehensive performance analysis of the algorithm as data scale and complexity increase, including potential optimization techniques for handling large datasets.

**Questions:**

1. **Why not use Deep Q-learning?**: The method used in the paper seems to have poor scalability and may only be suitable for relatively fixed knowledge systems. There is a lack of discussion on its scalability in large-scale educational environments.

2. **Sample Sizes in Datasets**: How were the sample sizes for the datasets set? In Junyi and OLI Statics, the quantity N is 100, but the number of topics varies. Is this N an empirical value or does it represent all the data in the original datasets? What is the actual N in this task, i.e., how many knowledge points should a model handle in practice? These aspects are not discussed in the main text, which is confusing.

3. **E(BI) Result of 100**: The paper reports an E(BI) result of 100 for the proposed method, but it is unclear what this result signifies compared to the random method. The paper also states that other results did not exceed 50, implying that these results are worse than random. Could you clarify why not exceeding 50 indicates worse performance than the random method and explain the significance of the E(BI) result of 100 in this context?

**Limitations:**

The authors have explicitly acknowledged the limitations of their work, providing explanations for these constraints and discussing their impact on the research. The paper does not pose any negative social impact, as it aims to improve educational practices through a more personalized approach to learning.

---

> ### Author Rebuttal · Authors · 2024-08-06
>
> We thank the reviewer for their feedback and aim to address them. We thank the reviewer for pointing out formatting errors and will clarify them in the future iteration.
>
> ## Questions
> 1. On simplified modelling:
> There seems to be a misunderstanding here. In our formulation of EdNetRMABs, arms only maintain 2 states - learned or unlearned. Our key distinction between classic RMABs and EdNetRMABS is the introduction of inter-related arms, and pulling an arm results in neighboring arms to be semi-active. When the actions have been selected, EdNetRMABs perform a rollout and neighboring arms will transit to the new state with $P(s’ | s, a = 1)$. We hope this clarifies any confusion.
>
> 2. On Bidirectional Relationships between knowledge points:
> While there are some earlier works regarding identifying pre-requisite and recommending educational content in the designed sequential order (e.g. Chang, Hsu & Chen, 2015), our work primarily focuses on topic groups and bidirectional relationships as a general case. In reality, it is difficult to isolate unidirectional relationships when learning education content: learning prerequisite content enhances learning in future content, while mastery over succeeding content may reinforce knowledge in prerequisite content. However, we do agree that learning advanced content without proper foundational knowledge will lead to the reverse learning effect that the reviewer described, and presents an interesting direction for future work.
> Finally, we wish to point out that EdNetRMABs can be easily extended to unidirectional structure, where $\phi_i \neq \phi_j$ for $j ∈ \phi_i$, and EduQate should operate similarly under such conditions.
>
> 3. On User Experience:
> We thank the reviewer for their suggestion regarding a user experience study, which we plan to leave for future work. In this work, we primarily focused on introducing a new RMAB and an approach to solve it, emphasizing theoretical credibility and rigor before implementing it in real-world settings.
>
> 4. On scalability issues, and why we did not choose Deep Q-Learning:
> Modeling such problems in RL is not trivial, as the state and action spaces grow combinatorially. If there are 100 arms (like in our real world datasets), then the state space is 2^100. The key advantage of RMABs is that they can do independent reasoning across the arms through Whittle index type approaches (thereby avoiding the significant combinatorial complexity).
> Deep Q-learning, while popular, suffers from the same issues that factored MDPs (Green et al (2011)) also suffers from. In both cases, the joint state space must be jointly considered for learning to take place. RMABs and EduQate however, can train in a decentralized settings, where update to each arm are independent.
>
> 5. Regarding sample sizes and how $N=100$:
> For both datasets, we selected the top 100 exercises with the most student interactions to create our student models (lines 271-272). This decision was made to ensure the development of realistic student models. For exercises with insufficient student interaction data, the resulting transition matrix was not meaningful. Specifically, some exercises were either solved on the first try by all students who attempted or never solved at all, resulting in an extreme transition matrix that is unlikely to represent real-world student behavior.
> For reference, N for the full Junyi and OLI Statics dataset were 837 and 300 respectively.
>
> 6. Regarding $IB$ as a metric:
> We provide the Equation for IB here for the reviewer's convenience.
> $IB_{Random,EQ}(\pi) = \frac{E_{\pi}(R(.)) - E_{Random}(R(.))} {E_{EQ}(R(.)) - E_{Random}(R(.))}$
> Note that when $\pi$ = EduQate, the numerator is equal to the denominator, resulting in 1.0. Our intention for $IB$ was a champion-challenger metric, where we compare EduQate (the champion model) versus the other baselines (challenger).
> In regard to the results, a negative value signifies that the baseline policy performs worst than the random policy, while a low positive value signifies extremely low performance when compared to EduQate. We acknowledge that the statement can be confusing, and have proposed an alternative definition of IB to ease the confusion: $$IB_{Random}(\pi) = \frac{E_{\pi}(R(.)) - E_{Random}(R(.))} {E_{Random}(R(.))}$$
> We have updated the results with the new definition in the PDF attached.

---

> ### Comment · Reviewer_fqUb · 2024-08-14
> **comment from reviewer**
>
> Thanks for the response.

---

> > ### Author Response · Authors · 2024-08-14
> >
> > Thanks for reading our response. It seems the reviewer lowered the score from 5 to 4. Can we please check if there was some issue with our response? Could we help clarify anything?

---

### Official Review · Reviewer_KG3r · 2024-07-12

**Soundness:** 2
**Presentation:** 2
**Contribution:** 2
**Rating:** 3
**Confidence:** 4

**Summary:**

The paper proposes EduQate, an innovative framework that leverages EdNetRMABs to achieve interdependence among knowledge points. By using Q-learning, EduQate implements optimal strategies for personalized learning, offering optimality guarantees without needing explicit knowledge of transition functions governing student learning states. This approach dynamically adapts educational content to individual progress, optimizing learning experiences and outcomes. The effectiveness of EduQate is validated using three real-world datasets, demonstrating its capability to identify and implement the most effective teaching strategies.

**Strengths:**

Overall, this article provides valuable insights and has a good overall quality. It has a certain impact on the AI education domain. The proposed approach, EduQate, adapts the Q-learning method of the multi-armed bandit model and introduces active and passive arms to establish correlations between knowledge points, which demonstrates a level of innovation. The article ensures the validity of the approach at the theoretical level, making it more persuasive. In terms of narrative, the article maintains logical coherence and is comprehensible. The discussed topics are indeed important in the field of education.

**Weaknesses:**

The formatting of the article is messy, and the images are not very clear. The layout on the sixth page is problematic, with formulas and text mixed together, making it confusing. The images on the seventh page and some of the appendix images are blurred. I suggest using a different format to redraw them.

There are too few baselines in the experimental section, and there are very few validation experiments. Merely presenting outstanding performance in one experimental metric is insufficient to demonstrate the superiority of your method. It would be beneficial to include some reinforcement learning baselines for comparison. Other reinforcement learning methods, such as the Bayesian network approach mentioned in the citations, can also accomplish the task. Therefore, the necessity of your method is unclear in the paper.

Your method does not consider the specific circumstances of practical problems. It raises fairness concerns. Ensuring that all knowledge points are fairly selected in a specific educational context is a crucial issue. Although the article presents a good framework, I believe it is not feasible to use it.

**Questions:**

Your claim seems to have ambiguity. In equation 5 of the proof, the expression for

$Q(s_i,a_i=1)-Q(s_i,a_i=0)+\sum_{j∈ϕ_i^-}(Q(s_j,a_j=1)-Q(s_j,a_j=0))$

However, in equation 1, the expression for λ_i is given as:

$Q(s_i,a_i=2)-Q(s_i,a_i=0)+\sum_{j∈ϕ_i^-}(Q(s_j,a_j=1)-Q(s_j,a_j=0))$

Did I miss something?

Regarding equation 9, I am unable to comprehend how $Q(s,a=I_i)$ is derived from the previous step without further information.

**Limitations:**

As mentioned above, there are many factors to consider in the specific field of education. These factors include fairness, mapping each question's knowledge points to the ARMs mentioned above, and the presence of cold-start problems, among others. Time is also a crucial factor to consider during real-time usage, although the paper does not mention specific algorithmic time information in practice.

---

> ### Author Rebuttal · Authors · 2024-08-06
>
> We thank the reviewer for the valuable feedback and the formatting concerns will be fixed.
>
> ## Questions
> 1. On implementing more RL baselines:
>  Please note that a key component of problems of interest are the presence of the arms (topics). Modeling such problems in RL is not trivial, as the state and action spaces grow combinatorially. If there are 100 arms (like in our real world datasets), then the state space is 2^100. The key advantage of RMABs is that they can do independent reasoning across the arms through Whittle index type approaches (thereby avoiding the significant combinatorial complexity).
> We discuss Green et al. (2011) in the related works section and justify the advantages of RMABs over factored MDPs (lines 95-99) with regards to the scalability. Furthermore, as supported by Reviewer 4CnK, bandits remain a core model behind today's learning platforms, which this work aims to enhance. In this work, we have devised a novel RMAB model, which can be leveraged upon by education settings for valuable personalized education.
>
> 2. On issues of fairness and ensuring all arms are fairly selected:
> We agree that the current work does not directly consider fairness across topics. We understand the reviewer's concern and looked through works on Fairness in RMABs that can be adapted to our setting. Please note that it is feasible to employ (with modifications) the work by Li and Varakantham (2023), citation number 14 to enable their proposed soft fairness constraint in EdNetRMABs.
>
> 3. On the claims and expressions:
> There is a typing error in Equation 5, where it should follow Equation 1:
> $\lambda_i = Q(s_i, a_i=2) - Q(s_i, a_i=0) + \sum_{j\in\phi_i^{-}}(Q(s_j, a_j=1) - Q(s_j, a_j=0))$
> We mistakenly replaced $Q(s_i, a_i=2)$ with $Q(s_i, a_i=1)$. We rewrite the Equation 5 to make it easier to read:
>
> $$\lambda_i \geq \lambda_j $$
> $$Q(s_i, a_i=2) - Q(s_i, a_i=0) + \sum_{p\in\phi_i^{-}}(Q(s_p, a_p=1) - Q(s_p, a_p=0)) \geq Q(s_j, a_j=2) - Q(s_j, a_j=0) + \sum_{q\in\phi_j^{-}}(Q(s_q, a_q=1) - Q(s_q, a_q=0))$$
> $$Q(s_i, a_i=2) + \sum_{p\in\phi_i^{-}}(Q(s_p, a_p=1))+Q(s_j, a_j=0)+ \sum_{q\in\phi_j^{-}}(Q(s_q, a_q=0)) \geq Q(s_j, a_j=2) + \sum_{q\in\phi_j^{-}}(Q(s_q, a_q=1))+ Q(s_i, a_i=0)+ \sum_{p\in\phi_i^{-}}(Q(s_p, a_p=0))$$
>
> * When arm $i$ and arm $j$ are not connected, but group $\phi_i$ and $\phi_j$ has overlap, i.e., $\phi_i \cap \phi_j \neq \emptyset $. In this case, we add $\underset{z\notin \phi_i \wedge z\notin \phi_j}{\sum} Q(s_z, a_z=0) - \sum_{z\in \phi_i \cap \phi_j} Q(s_z, a_z=0)$  on both sides, we can have the left side(Equation 9):
>
> $$Q(s_i, a_i=2) + \sum_{p\in\phi_i^{-}}(Q(s_p, a_p=1))+Q(s_j, a_j=0)+ \sum_{q\in\phi_j^{-}}Q(s_q, a_q=0) + \underset{z\notin \phi_i \wedge z\notin \phi_j }{\sum} Q(s_z, a_z=0) - \sum_{z\in \phi_i \cap \phi_j} Q(s_z, a_z=0)$$
> $$ = Q(s_i, a_i=2) + \sum_{p\in\phi_i^{-}}(Q(s_p, a_p=1))+\ \sum_{q\in\phi_j}Q(s_q, a_q=0) + \underset{z\notin \phi_i \wedge z\notin \phi_j }{\sum} Q(s_z, a_z=0) - \sum_{z\in \phi_i \cap \phi_j} Q(s_z, a_z=0) (\text{Combined the third with the fourth term})$$
> $$= Q(s_i, a_i=2) + \sum_{p\in\phi_i^{-}}(Q(s_p, a_p=1))+\sum_{q\notin\phi_i}(Q(s_q, a_q=0)) (\text{Combined the last three terms})$$
> $$= Q(\textbf{s},\textbf{a}=\mathbb{I}_{ \{ i=2 \} })$$
>
> We revised $Q(\textbf{s},\textbf{a}=\mathbb{I}_{i})$
>
> to $Q(\textbf{s},\textbf{a}=\mathbb{I}_{\{i=2\}})$ to more clearly denote the action of selecting only arm $i$.
> For brevity, we do not include the full proof here, and will correct the errors in future iteration.
>
> 5. On limitations of mapping knowledge points, cold-start problems, and time factors:
> In this work, we have addressed several limitations of our method, particularly the cold-start problem, for which we have attempted to circumvent by proposing Experience Replay (lines 185-191). An ablation study in Appendix E.2 demonstrates that EduQate achieves reasonably good results in fewer than 100 episodes.
> Next, regarding methods to derive groupings for knowledge points, we briefly discuss data mining methods in lines 100-104, which focus on deriving relationships between learning content. Our work leverages these works to create the EdNetRMABs and to enhance the learning experience of students.
> Finally, once Q-learning converges and is applied in the real world, it is equivalent to a hashmap retrieval operation with a time complexity of O(1). In this regard, we do not think that the time complexity of EduQate will be a major issue in the learning experience of the student. The number of episodes required for Q-learning convergence is discussed in Appendix E.2 as well.

---

### Official Review · Reviewer_Yn9L · 2024-07-13

**Soundness:** 3
**Presentation:** 4
**Contribution:** 3
**Rating:** 8
**Confidence:** 3

**Summary:**

This paper proposes a solution to generate personalized learning curricula in educational settings, focusing on the challenge of accounting for interdependencies between learning topics. It argues that existing approaches, often based on the Restless Multi-Armed Bandit (RMAB) framework, fall short by assuming independence of learning content which is unrealistic in educational settings. To address this the paper introduces a new model Restless Multi-armed Bandits for Education (EdNetRMABS) enabling the capture of relationships between independent learning items. Building upon this model, the authors propose a new algorithm called EduQate, which leverages Q-learning and the Whittle index to compute an interdependency-aware teacher policy for recommending educational content. Notably, EduQate doesn't require prior knowledge of the transition matrix, unlike traditional Whittle index methods.

The authors provide a theoretical analysis demonstrating the optimality of EduQate for the case of recommending a single item at each time step (k=1). While finding the optimal solution for recommending multiple items (k>1) is proven to be NP-hard, a heuristic greedy algorithm is proposed to find solutions.

Through experiments on synthetic and real-world datasets (Junyi and OLI), the paper demonstrates the superiority of EduQate over baseline policies, including Threshold Whittle (TW), WIQL, Myopic, and Random. EduQate consistently achieves higher intervention benefits and average rewards across all datasets. Further analysis reveals the effectiveness of the replay buffer in EduQate, mitigating the "cold-start problem" common in reinforcement learning applications.

**Strengths:**

1. The paper is well organized and very well written with comprehensive explanations of the model, algorithm and the experimental setup.
2. Addressing the crucial issue of interdependency in learning content is a significant contribution. EdNetRMABs offer a realistic model for educational settings, moving beyond the simplifying assumption of independence prevalent in traditional RMAB approaches.
3. The paper presents a rigorous theoretical analysis of EduQate, proving its optimality for the k=1 case and providing complexity bounds for the k>1 case.
4. The authors address the cold-start problem by incorporating experience replay and perform ablations to show its effectiveness. This is a practical enhancement relevant for real-world applications.

Overall, this paper presents a strong contribution to the field of adaptive learning by introducing a novel and effective approach for generating personalized curricula that account for interdependencies in learning content.  The theoretical analysis, empirical results, and focus on practical considerations make this work both insightful and impactful.

**Weaknesses:**

The assumption of fully observable knowledge states is a significant limitation. Future work should explore extending the model and algorithm to handle partial observability, a more realistic scenario in education.
While the complexity analysis is provided, further investigation on the scalability of EduQate to larger datasets and more complex networks would strengthen its practical applicability.
The current work primarily focuses on maximizing long-term rewards. Further analysis on balancing exploration and exploitation in EdNetRMABs could offer valuable insights for curriculum design.

**Questions:**

Can EduQate's recommendations be easily interpreted?  Understanding the rationale behind recommendations can enhance trust and facilitate pedagogical insights.
Are there any potential ethical implications of using EdNetRMABs and EduQate in education?

**Limitations:**

1. The experiments are based on simulated students and existing datasets. Real-world classrooms involve numerous factors not accounted for in the model, such as student motivation, engagement or diverse learning styles. This may limit the practical utility of this method.
2. The paper does not address the interpretability of EduQate's recommendations.
3. The assumption that the student states are fully observable is a major limitation of this work. This can result in overconfident recommendations.

---

> ### Author Rebuttal · Authors · 2024-08-06
>
> We thank the reviewer for the positive feedback and comments. We address some of their questions and concerns below:
>
> ## Questions
> 1. Can EduQate's recommendations be easily interpreted?
> Reviewers point on interpretability is well noted. Given that RMABs are typically simple models, where arms have only few states (good progress, bad progress etc.) and Whittle index indicates the potential for improvement on that arm, they are very amenable to providing interpretations. For these interpretations to be deeper and meaningful, we also need to explain why the Whittle index takes on a certain value . We can potentially build from work on interpretability in MDPs, but unfortunately, this is non-trivial and difficult for us to show results before the end of the rebuttal period.
>
> 2. Are there any potential ethical implications of using EduRMABs and EduQate in education?
> The current EdNetRMABs require the specification of inter-item relationships. Without a learning-based method or end-to-end algorithm to learn these relationships and inform EdNetRMABs, it is possible for bad actors to withhold exposure to certain educational materials from specific groups.
> As with other educational tools, EdNetRMABs are intended to be applied to any domain and thus could potentially be used to impart undesirable skills (e.g., terrorism). Therefore, EdNetRMABs still require careful curation of learning content.
>
> 3. The assumption that the student's state is fully observable is a major limitation of the work. This can result in overconfident recommendations.
> We acknowledge the issue of full observability and plan to address it in our future work. Despite this limitation, our results have shown promise over existing methods.

---

> ### Comment · Reviewer_Yn9L · 2024-08-11
> **Acknowledgement of rebuttal**
>
> Thank you for addressing my questions.

---

### Official Review · Reviewer_4CnK · 2024-07-14

**Soundness:** 4
**Presentation:** 4
**Contribution:** 3
**Rating:** 8
**Confidence:** 4

**Summary:**

This paper presents a way to extend RMAB with Q-learning to accommodate the fact that some items/arms belong to a group. RMAB can be considered as a weakened version of contextual bandit CB but also a strong version of CB since it considers state transitions (explicitly defined on arms).

**Strengths:**

This is a very solid paper with quite a few reasons to accept it:
1. The contribution of education simulators. Can the authors comment on whether they will release their simulators publicly for other education or bandit communities to use in the future? Currently, there is a lack of high-quality education simulators that are based on real-world data. This could be a great contribution. I hope the authors make an effort to make such resources public.
2. The technical contribution of introducing a pseudo-action that extends RMAB and WIQL.
3. Clean writing and presentation.

MAB has been used in many different real-world settings and is the main algorithm behind many learning platforms. Any innovation in this space will have a huge impact on students around the world. Unfortunately, this area is very niche and requires high technical sophistication. Unless someone can convince me that this paper's algorithm has already been published elsewhere or is fully derivative of some other work, I think it's a great paper to present at NeurIPS.

I might also be a bit concerned if the authors decide not to share their code/algorithm/simulators to encourage more future research in this direction.

**Weaknesses:**

1. Can add some more ablation studies.
2. Some clarifications (see question section).

**Questions:**

I have a few questions related to the technical contributions of the paper. None of these questions are reasons for rejection. I just want some clarification from the authors.
1. Equation (1) -- it seems to me that this altered the original definition of Whittle Index by summing over pseudo-actions. Can you explain by doing so, would it change the optimality of Whittle Index as a solution to RMAB? For example, how do we know under such an extension, the Whittle Index can still get the optimal solution? Is Theorem 1 answering this question? If so, does the original Whittle Index algorithm guarantee optimality (I assume over sublinear regret?) for both k=1 and k>1 arm selection strategy? I apologize if my understanding of RMAB is inaccurate.
2. If it's easy to run, can you add an ablation study that's under misspecification of $\phi$? Often in education, the grouping of items/materials is not exact -- and can often be wrong. The membership function $\phi$ has errors. I'm a bit curious about how much the quality of $\phi$ impacts EduQate. I understand it's hard to add experiments given the tight rebuttal period, so don't worry if this is too much of an ask.
3. Eq 4 shows IB as a metric. My question is: if both $\pi$ and $EQ$ are worse than random -- then IB will still be positive. This is not a metric that shows "successful algorithms must demonstrate superiority over random policies" -- this is a metric that says we should have a policy better than the baseline, but it can be worse. Do I understand this correctly? IB is a valid metric to use. Also, why is IB 100% in Table  1? Maybe my intuition is just off, but how is 100% achieved? Can you give an example?

**Limitations:**

The authors discussed limitations in a section.

---

> ### Author Rebuttal · Authors · 2024-08-06
>
> We thank the reviewer for his positive feedback and comments, and address their questions below:
>
> ##Questions
> 1. On releasing code/simulators
>     We will share our code for community use.
> 2. On Equation (1): Does summing over the pseudo-actions change the optimality of Whittle Index? Is Theorem 1 answering the question?
>     Whittle index is optimal only under some strict assumptions/requirements. One such fundamental requirement is that each arm must be indexable. Indexability is characterized by the property where the set P(λ), representing all states in which deactivating the arm is optimal at a game cost λ, must increase monotonically from the empty set to encompass the entire state space as λ increases from 0 to +∞. In our model, the arms exhibit interdependencies, meaning that the index value of any given arm can be influenced by the states of its neighboring arms. Thus, they complicate the computation of the traditional Whittle index. The decision to activate an arm can become more or less favorable depending on the states of other interconnected arms, not solely based on its own state and the subsidy λ. Thus, this summing over pseudo-actions impacts optimality.
> Our theorem 1 demonstrates that despite these complexities, it is possible to adapt the Whittle index to account for interdependencies among arms. This adaptation leverages the inherent structure of interdependencies to refine the decision-making process, thereby aligning it with the theoretical framework required for the optimality of the Whittle index policy.
> 3. On ablation study under misspecification of $\phi$:
> Please note that $\phi$ represents inter-item relationships and this is usually not very complex or extensive. We believe that trainers/teachers can quickly verify the specified relationships to remove any errors.
> However, we understand the reviewer's concern. We hypothesise that misspecification of $\phi$ will result in erroneous attribution of positive effects to passive actions and negative effects to pseudo-actions for EduQate. We provide a trial run of the conditions where a proportion of $\phi$ is misspecified in Table 2 of the attached PDF. In this case, EduQate outperforms random policy and WIQL. Even without knowing the underlying transition matrix, EduQate is still able to perform as well as TW (which requires knowledge of transition probabilities).
> Additionally, WIQL can be seen as a special case of EduQate where $\phi_i = \emptyset$ for all $i$, representing an extreme case of misspecification.
>
> 4. On $IB$ as a metric:
> The reviewer is correct on the interpretation of $IB$ and we agree with the reviewer's concern when random performs better. IB has been used in RMABs in healthcare settings, albeit over a no-action policy as the baseline. We adapt it to education settings and compare our policies to a random policy, which is known to be a strong policy for education. We agree that in certain cases where random outperforms both $\pi$ and EduQate, $IB$ can be misleading. In our main experiments, $R_{EQ} > R_{random}$ for all cases, and hence the current version of $IB$ is valid.
> We propose to modify $IB$ to $$IB_{Random}(\pi) = \frac{E_{\pi}(R(.)) - E_{Random}(R(.))}{ E_{Random}(R(.))}$$ for simplicity, while maintaining the original intention of comparing it to the random policy baseline. We present the updated results in Table 1 of the attached PDF.
> For EduQate achieving 100%, we present the following. Let $\pi = EQ$, then:
> $$IB_{Random,EQ}(EQ) = \frac{E_{EQ}(R(.)) - E_{Random}(R(.))} {E_{EQ}(R(.)) - E_{Random}(R(.))} = 1$$

---

> ### Comment · Reviewer_4CnK · 2024-08-11
>
> The new metric seems much more reasonable! Thanks for the explanation.
>
> I read through the rejection reviews as well. Multi-Arm Bandit is still one of the most commonly used algorithms by education platforms such as Duolingo and others. As far as bandit paper goes, this paper's experimental results are sufficient.
>
> I will keep my score, and I'm happy to defend this paper and my decisions.

---

### Author Rebuttal · Authors · 2024-08-06

We thank the reviewers for their time and positive feedback! We are pleased to know that the reviewers found our work significant and an important contribution to the field of education technology (Reviewers 4CnK, Yn9L, fqUb, KG3r).

We wish to point out that the IB metric is not introduced by us but is a well accepted metric in RMAB literature. However, reviewers' concerns are well noted. In addition to the current IB metric, we propose a modification to IB that we believe can address the concerns of the reviewers:
$$IB_{Random}(π) = \frac{E_π(R(.)) - E_{Random}(R(.))}{E_{Random}(R(.))}$$

We present the results with this new metric in Table 1 of the attached PDF. All feedback will be incorporated into the updated manuscript.

---

### Decision · Program_Chairs · 2024-09-25

**Decision:**

Reject

**Comment:**

The reviewers for this paper had a diverging assessment with a huge spread in the ratings (8, 8, 4, 3, 3). On the one hand, several reviewers had a positive assessment, highlighting the paper's strong contributions of introducing a new approach for creating personalized curricula that can account for interdependencies between items. On the other hand, several reviewers had negative assessment, raising concerns related to limited experimental evaluation and modeling that might not capture real-world educational settings in practice. We want to thank the authors for their detailed responses. Based on the raised concerns and follow-up discussions, unfortunately, the final decision is a rejection. Nevertheless, the reviewers have provided detailed and constructive feedback. We hope the authors can incorporate this feedback when preparing future revisions of the paper.